# Seeking diagnostic and prognostic biomarkers for childhood bacterial pneumonia in sub-Saharan Africa: study protocol for an observational study

Clarissa Valim ![ORCID],[1] Yekin Ajauoi Olatunji,[2] Yasir Shitu Isa,[2] Rasheed Salaudeen,[2] Sarwar Golam,[2] Edward F Knol,[3] Sheriffo Kanyi,[4] Abdoulie Jammeh,[5] Quique Bassat,[6,7] Wilco de Jager,[3,8] Alejandro A Diaz,[9,10] Roger C Wiegand,[11] Julio Ramirez,[12] Marsha A Moses,[13,14] Umberto D'Alessandro,[15,16] Patricia L Hibberd,[17] Grant A Mackenzie ![ORCID] [2,18]

PLH and GAM contributed equally.

**Correspondence to**
Dr Clarissa Valim;
cvalim@bu.edu

## ABSTRACT

**Introduction** Clinically diagnosed pneumonia in children is a leading cause of paediatric hospitalisation and mortality. The aetiology is usually bacterial or viral, but malaria can cause a syndrome indistinguishable from clinical pneumonia. There is no method with high sensitivity to detect a bacterial infection in these patients and, as result, antibiotics are frequently overprescribed. Conversely, unrecognised concomitant bacterial infection in patients with malarial infections occur with omission of antibiotic therapy from patients with bacterial infections. Previously, we identified two combinations of blood proteins with 96% sensitivity and 86% specificity for detecting bacterial disease. The current project aimed to validate and improve these combinations by evaluating additional biomarkers in paediatric patients with clinical pneumonia. Our goal was to describe combinations of a limited number of proteins with high sensitivity and specificity for bacterial infection to be incorporated in future point-of-care tests. Furthermore, we seek to explore signatures to prognosticate clinical pneumonia.

**Methods and analysis** Patients (n=900) aged 2–59 months presenting with clinical pneumonia at two Gambian hospitals will be enrolled and classified according to criteria for definitive bacterial aetiology (based on microbiological tests and chest radiographs). We will measure proteins at admission using Luminex-based immunoassays in 90 children with definitive and 160 with probable bacterial aetiology, and 160 children classified according to the prognosis of their disease. Previously identified diagnostic signatures will be assessed through accuracy measures. Moreover, we will seek new diagnostic and prognostic signatures through machine learning methods, including support vector machine, penalised regression and classification trees.

**Ethics and dissemination** Ethics approval has been obtained from the Gambia Government/Medical Research Council Unit The Gambia Joint Ethics Committee (protocol 1616) and the institutional review board of Boston University Medical Centre (STUDY00000958). Study results will be disseminated to the staff of the study hospitals, in scientific seminars and meetings, and in publications.

## Strengths and limitations of this study

► This study will be conducted in a resource-limited country for future development of a point-of-care test for these settings.

► We will quantify known and recently identified biomarkers that will be combined using different machine learning approaches.

► Children will be followed up for 30 days after discharge to yield mid-term prognostic assessment of the clinical pneumonia.

► A difficulty shared by studies aiming to identify biomarkers to diagnose bacterial infection is lack of an optimal gold standard for bacterial infection; thus, we will use a composite criterion based on microbiological and radiological evidence of infection (primary criteria) and probable criteria based on expert evaluation of clinical and laboratory data.

► Although the sample size is relatively small, results will establish a benchmark against which the identified signatures in populations in other areas and in different subgroups, such as children with undernutrition, can be compared.

**Trial registration number** H-38462.

## INTRODUCTION

Clinical pneumonia may be caused by viral or bacterial infections[1] and, where malaria is endemic, malaria can cause a syndrome indistinguishable from clinical pneumonia.[2] There has been a growing number of publications on biomarkers for the diagnosis of bacterial infections and to prognosticate outcomes in patients with clinical pneumonia.[3–37] Such increased attention is driven by the need for low cost point-of-care (POC) tests able to accurately, reliably and promptly diagnose bacterial infections that can guide

treatment.[38–41] All existing tests have delayed turnaround times or require specialised facilities that are often unavailable in resource-limited settings. POC tests can support timely decisions about treatment and ameliorate the overprescription of antibiotics to treat viral infections, thus slowing the progression of global antibiotic resistance.[42–46] In malaria endemic areas, antibiotics may also be 'underprescribed' and children with bacterial pneumonia may be mistakenly treated for malaria and not receive antibiotic therapy.[43 47] Prognostic POCs could also guide referrals and allow prioritisation of intensive treatment measures that are particularly scarce in resource-limited settings.

Symptoms of bacterial and viral diseases and of malaria often overlap.[48 49] Radiographical evidence of pneumonia has been used as an endpoint in studies of pneumococcal vaccine, but chest X-rays, particularly in resource-limited settings, when available have moderate reliability[50–58] and may result in both false-positive and false-negative results. Laboratory tests have low accuracy. Blood or pleural fluid cultures are highly specific for diagnosing bacterial infections but have low sensitivity, and results are not promptly available.[59 60] Bacterial PCR tests require specialised resources and have low sensitivity. Viral PCR also requires specialised facilities and does not establish a diagnosis of viral pneumonia.[49 61–63] Furthermore, in children, antigen detection in the urine has limited utility, and induced sputum samples are difficult to obtain and likely to be contaminated with the normal flora.[64–67] Where malaria is endemic, standard malaria diagnostic tests can detect the parasite in blood. Nevertheless, malaria may coexist with other infections while not causing the respiratory syndrome.[68 69]

Generally, previous biomarker studies to diagnose bacterial diseases or to prognosticate disease in patients with pneumonia or sepsis focused on procalcitonin (PCT) and/or C reactive protein (CRP) and concluded that the markers had moderate accuracy (area under the receiver operating characteristic curve (AUC-ROC)<0.80).[9–14 34–36 70–93] Accuracy of a single biomarker in bacterial pneumonia is probably low since many biomarkers may be in the inflammatory pathway triggered by several pathogens.[94] More recently, studies screened through novel biomarkers and identified gene transcription signatures able to distinguish patients with bacterial and viral infections.[95 96] Additionally, a biomarker signature based on haptoglobin (HP) and lipocalin (NGAL) has been identified to differentiate malaria and probable bacteria aetiology in paediatric patients with clinical pneumonia in a resource-limited country.[97] Two other promising signatures to differentiate viral from bacterial infections in high-income countries (HICs) have been identified. The first (ImmunoXpert) is based on CRP, tumour necrosis factor-related apoptosis-inducing ligand (TRAIL) and interferon gamma-induced protein 10 (IP-10) and has been validated in HICs, with sensitivity ranging from 87% to 94% and specificity from 90% to 94%.[5 98–101] The second is based on CRP and Mixovirus Resistance Protein A and has been implemented in a POC (FebriDx) in HICs, with sensitivity and specificity varying from 80% to 96% and from 67% to 94%, respectively.[4 102 103] Although promising, these two signatures need to be validated in resource-limited settings where inflammatory responses may vary because of malaria, coinfections and undernutrition.[104]

In Mozambique, we scanned through several biomarkers and combined them through machine learning methods, aiming to detect bacterial infection (vs no bacterial infection) in paediatric patients with clinical pneumonia. Considering the potential of coexisting malaria infections, we sought biomarker signatures by comparing children with malaria, bacterial and viral infections and validated accurate signatures by comparing patients with and without bacterial infections.[105] Signatures with 3–5 proteins were identified that had high sensitivity for the diagnosis of bacterial pneumonia (96%–91%) while misdiagnosing few viral cases. The selected signatures performed better than signatures including commonly collected clinical and laboratory markers, as well as pneumonia with consolidation on X-ray. The current study builds on our previous findings and aims to validate previous diagnostic signatures identified by us and other groups, in addition to improving on signatures by scanning through novel biomarkers in a different population in rural Gambia.

Biomarkers for signatures will be selected and combined using a comprehensive set of data mining approaches. To validate and improve on prior signatures and seek prognostic signatures, we will add some novel promising biomarkers such as resistin and lactoferrin.[37 106 107] At the end of this study, we expect to select signatures with excellent accuracy (≥93% sensitivity) that have the highest prognostic utility and composed of a limited number of biomarkers that can be developed in a POC test. Our focus is on identifying patients who need antibiotic therapy for bacterial diseases.

## Study objectives

Specifically, this study aimed to

► Assess combinations of blood proteins to accurately and reliably diagnose bacterial pneumonia in paediatric patients.

We expect that protein combinations that were identified in previous studies as accurate for diagnosing bacterial infection in patients with clinical pneumonia will have high sensitivity, specificity and AUC-ROC in an African paediatric population different from the Mozambican children. Biomarkers of these signatures will be absent in healthy children. Moreover, we hypothesise that accuracy of previous signatures can be improved by seeking to combine them with novel candidate proteins.

► Explore prognostic biomarker signatures in patients with clinical pneumonia caused by any pathogen.

We anticipate that combination of inflammatory proteins in the blood measured at admission will accurately identify patients who will clinically deteriorate.

## METHODS AND ANALYSIS
### Design
To test and identify existing and novel diagnostic biomarker signatures for bacterial disease, we will measure proteins in the blood at admission and collect clinical and laboratory information at admission to ascertain diagnostic groups. Diagnostic signatures will be sought by comparing children in the target diagnosis groups. To seek prognostic biomarker signatures, we will prospectively follow clinical outcomes collected after admission and discharge. When studying prognostic signatures in the primary analysis, proteins will be compared across three groups with poor, moderate and good prognoses. The study started in February of 2019 and is expected to be concluded by November 2021.

### Study environment
The study will be conducted at the Boston University School of Public Health (BUSPH), the Medical Research Council Unit The Gambia (MRCG) at London School of Hygiene & Tropical Medicine, and the Multiplex Core Facility of the Centre of Translational Immunology from the University Medical Centre Utrecht. Patients will be recruited at Basse and Bansang Hospitals. These hospitals serve a population of 178 510 (224 villages) and 99 113 (217 villages), respectively, with 19% of the population aged <5 years. Diagnostic laboratory tests will be conducted at the MRCG, Basse Field Station. The multiplex bead-based immunoassay will be conducted at the Center for Translational Immunology/Utrecht Medical Center and analysis will be conducted at BUSPH.

### Outcomes
Two sets of outcomes will be evaluated, one for each study aim.

#### Objective 1: diagnostic group classification
The primary objective of the diagnostic group classification was to identify patients in need of antibiotic therapy for a bacterial infection and not to identify the pathogen responsible for the respiratory symptoms. Patients will be assigned to bacterial, malarial and viral groups according to a definitive classification (primary analysis) (figure 1) when seeking signatures, and bacterial and non-bacterial infection groups when testing signatures. Patients will also be categorised into bacterial and non-bacterial infection groups based on a probable classification criterion (secondary analysis). Patients with ambiguous diagnosis who do not meet our criteria will be excluded from the study of objective 1. By incorporating the two outcomes, we will assess the validity of the identified biomarker signatures in patients with and without detected bacteraemia and provide data to support the generalisability of the inflammatory biomarker signature.

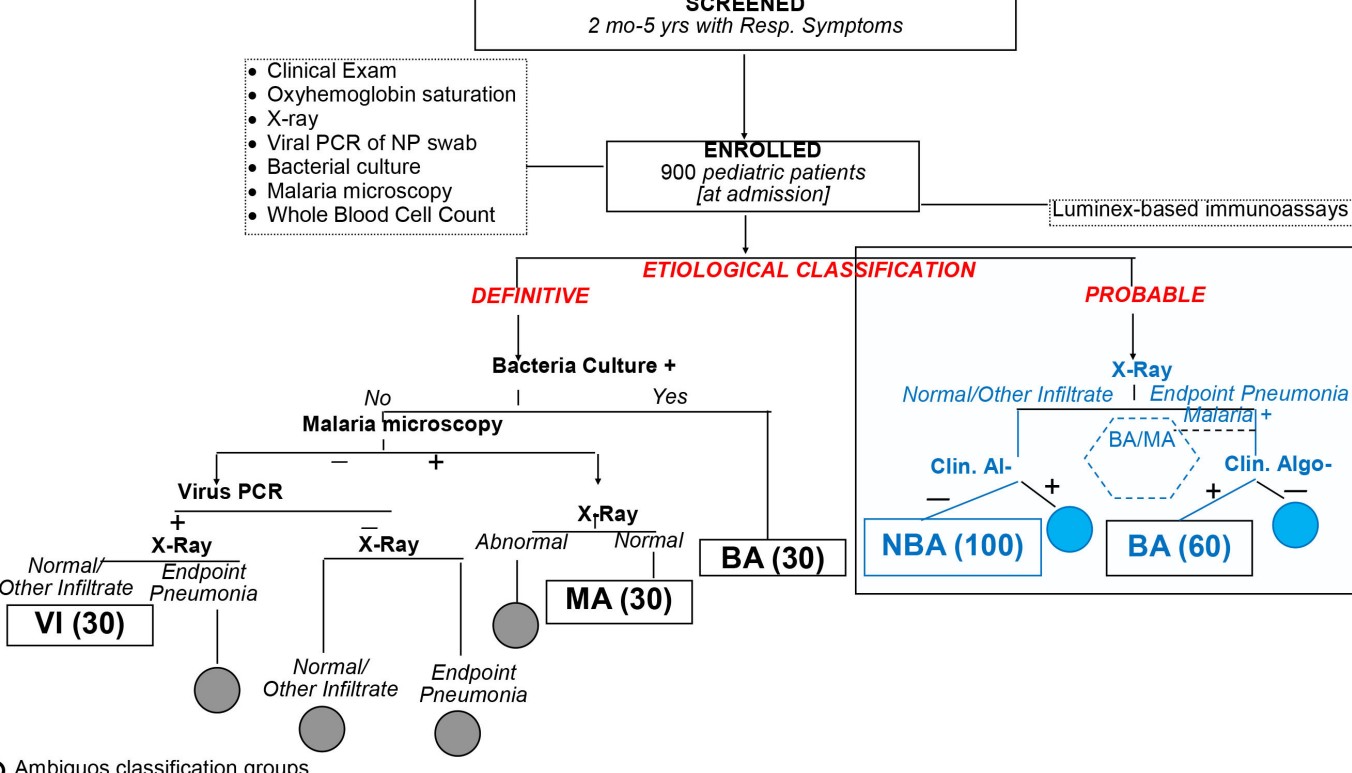

**Figure 1** Definitive and probable diagnostic classification criteria (gold standard of diagnostic signature aim) to assign patients to the MA, BA and VI groups (definitive composite criteria) and to the BA and NBA groups. BA, bacterial; MA, malaria; NBA, non-bacterial; NP, nasopharyngeal; VI, viral.

For the definitive classification, a patient will be assigned to the bacterial (BA) group if he/she has a positive bacterial pathogen culture of fluid from a normally sterile site (eg, blood or pleural fluid). Patients will be assigned to the viral (VI) group if they have negative bacteria microbiological tests, negative malaria blood slides, X-ray without 'endpoint pneumonia' (consolidation or pleural effusion[51]), no evidence of fungal infection, and positive PCR for a viral pathogen from nasopharyngeal swabs (NPS). Patients will be assigned to the malarial (MA) group if they have normal X-ray (with neither infiltrates nor endpoint pneumonia), no bacterial infection and >0/µL asexual *Plasmodium falciparum* parasites if they are aged <1 year or >2500 asexual parasites/µL of blood if they are aged >1 year. Patients who are admitted with viral infections but who develop a secondary bacterial pneumonia during hospitalisation will be excluded from the VI group.

Patients with HIV infection or exposure (<1 year old) will be categorised as fungal pneumonia if they have chest X-ray with bilateral/symmetrical reticular or granular opacities without lobar consolidation and severe hypoxia (≤85% at admission) or positive NPS PCR for *Pneumocystis jiroveci* or a blood culture positive to any pathogenic fungal micro-organism.

For the diagnostic categorisation of probable bacterial pneumonia, patients will be assigned to the *probable BA* group based on review of data by an expert panel including paediatricians and infectious diseases specialists with at least 10 years of experience. Data will include clinical information collected at and during admission, chest X-ray results, full blood cell counts (FBCs), malaria microscopy and/or rapid diagnostic test (mRDT), bacteria culture, viral NPS, and other laboratory test results, when available, for example, urinalysis. Two experts will review patient data, independently and blindly, classifying patients as 'certainly' BA or non-bacterial (NBA), 'probable' BA or NBA, 'unclear' BA or NBA. Only patients with consensual certainly or probable diagnosis will be included in this analysis.

## Objective 2: poor prognosis

Poor prognosis will be primarily based on a three-category outcome recorded during and after hospitalisation. Additionally, we will explore secondary outcomes associated with a poor evolution of the clinical pneumonia during the first 5 days of hospitalisation, at discharge and 30 days after admission.

For the *three-category outcome*, regardless of the original infection, patients will be selected from the following three groups: group a, children who die during or up to 30 days after admission; group b, children with prolonged hospital stay (>7 days) or who need to have antibiotic therapy changed within 48 hours of admission, or who are readmitted within 30 days after the first admission with symptoms associated with the first admission; and group c, children discharged well within 3 days of admission and without the need for a change in antibiotic therapy after

admission. Secondary outcomes of clinical deterioration will be (1) area under the first 5 days of the oxygen saturation curve; (2) necessity to change antibiotic therapy within the first 2 days from admission or to receive oxygen therapy when it was not required at admission and (3) duration of hospital admission.

## Participants

We will enrol 900 patients with clinical pneumonia aged 2–59 months by January 2021, assuming we will have approximately 30 patients with a definitive diagnosis of bacterial infection, 30 with malaria and 30 with viral infections (figure 1). Among the 900 patients, we anticipate enrolling 60 and 100 patients with and without probable bacterial infection, respectively.

For the study of prognostic biomarkers (figure 2), we will include 160 patients. All deaths during or up to 30 days from admission will be selected, and the remaining sample size will be divided between the other two category outcomes, that is, group b, prolonged admission, readmission or changed antibiotics during admission, and group c, discharged within 3 days with improvement of symptoms. To meet the target sample size in the groups b and c, eligible children will be randomly selected.

A total of 20 community healthy control children aged 2–59 months without any symptoms or signs and no malaria infection will be enrolled in vaccination clinics in Basse to assess precision of signatures.

## Patient and public involvement

Patients and the public were not involved in the design of this research.

## Inclusion and exclusion criteria

Paediatric patients aged 2–59 months will be screened and will be enrolled if they present with cough or difficult breathing, the guardians provide informed consent, are referred for probable admission, and have at least one of the following signs: increased respiratory rate for age based on the WHO criteria, lower chest indrawing, peripheral arterial oxygen saturation <93% (measured by pulse oximetry), grunting, nasal flaring or undernutrition. The WHO criteria for increased respiratory rate are respiratory rate ≥50 breaths/min for children 2–11 months old; and respiratory rate ≥40 breaths/min for children 1–5 years old.[108] Undernutrition will be defined as visible wasting or middle upper arm circumference <11.5 cm if aged ≥6 months or weight-for-height Z-scores of <−3. Patients will be excluded if they have suspected tuberculosis based on a cough lasting >2 weeks, were admitted to a hospital in the previous 2 weeks and have evidence of any condition that could be worsened by collection of blood.

Healthy community controls will be enrolled if an mRDT is negative; vaccines have not been administered that day; there are no symptoms of any disease; and there is no history of hospital admission in the previous 4 weeks. Healthy controls will be excluded if they are later found to have malaria infection detected by a positive microscopy.

| 3 PROGNOSIS GROUPS (N = 160) | | |
|---|---|---|
| **Poor prognosis** | **Adverse progress of disease** | **Good Prognosis** |
| • Dearth | • Re-admission<br><br>• Need to switch antibiotic therapy ≤ 48 hrs of hospitalization<br><br>• > 3 days of hospitalization | • Discharged well<br><br>• No need to switch antibiotic therapy during admission<br><br>• < 3 days of hospitaliza- |

| CONTINUOUS AND CATEGORICAL POOR PROGNOSIS OUTCOMES (N = 160) |
|---|
| • Area under 5-day $SaO_2$ |
| • Need to switch antibiotic therapy ≤ 48 hrs of hospitalization |
| • Length of hospital stay |

**Figure 2** Exploratory primary (A) and secondary (B) prognosis outcomes.

### Recruitment and follow-up

On arrival at any of the study hospitals, patients aged 2–59 months will be screened for the presence of respiratory symptoms and undernutrition by a senior research nurse with extensive experience in research in paediatric pneumonia (figure 1). If they pass this first-level assessment, they will be referred to a second research nurse for a detailed medical history and physical examination. Those patients who are considered eligible will undergo informed consent procedures.

After informed consent, patients will be referred for clinical examination and admission procedures by a research clinician and then for sample collection and chest X-ray. NPS will be performed and 5 mL of venous blood will be drawn (never more than 3 mL/kg per WHO standard[109] with 3 mL immediately placed into one paediatric blood culture bottle (Pedibact; BD, Franklin Lakes, New Jersey, USA) and the remaining 2 mL placed in an EDTA vacutainer. From the EDTA tube, a drop of blood will be used for haemoglobin rapid test and mRDT, both part of standard of care in the Gambia. Other tests based on standard of care may be conducted, including urinalyses, glucose, examination of cerebrospinal fluid (CSF) and pleural fluid, or lung aspirate. We will follow local clinical practice and HIV testing will not be routinely conducted. HIV prevalence is approximately 1% in antenatal surveys in The Gambia. Blood in EDTA will be processed and used within 1 hour from collection for FBC, malaria microscopy, and will be centrifuged and aliquoted to store plasma at −80C in the MRCG biobank for the multiplex bead-based immunoassay. An aliquot of 150 μL will be sent to the Centre of Translational Immunology in Utrecht for quantification of proteins with temperature controlled dry ice shipment.

### Follow-up

After admission, patients will be followed up daily for 5 days to record pulse oximetry, antibiotic therapy, feeding, clinical deterioration with need for additional tests. If patients are receiving oxygen during the follow-up visits, they will have their lowest recorded pulse oximetry imputed. At hospital separation, a research nurse will fill a case report form (CRF) to record diagnosis and outcome (eg, death, absconding or discharge alive and well). Moreover, for patients who are discharged, a phone call will be placed to the parent/guardian 30 days after admission.

### Study measurements and procedures

► Bacterial culture: approximately 3 mL will be incubated in an automatic BACTEC 9050 system (BD) for a minimum of 4 days. Positive blood cultures will be examined by Gram stain and subcultured onto blood agar, chocolate agar or MacConkey agar plates based on Gram stain. Bacterial growth will be defined as contaminant when bacteria generally considered skin flora are isolated from the positive blood culture (eg, coagulase-negative Staphylococci, *Bacillus* species or Micrococci).[110] Lung fluids will be cultured according to procedures previously described.[111]

► Culture of CSF, lung aspirate and pleural fluid aspirate: cell count and direct gram reaction will be performed as well as bacterial antigen latex agglutination test. A 50–100 µL aliquot of the specimen will be inoculated onto blood agar, chocolate agar and MacConkey Agar plates for isolation of bacterial growth.

► Viral PCR of NPS: NPS will be placed in media that will be aliquoted and stored at −80°C for real time polymerase chain reaction (rtPCR). Assays will use primers and probes for influenza virus types A and B, respiratory syncytial virus types A and B, adenovirus, parainfluenza viruses 1, 2, 3 and 4, human metapneumovirus, coronavirus, bocavirus, swine influenza virus and rhinovirus. *P. jiroveci* will also be screened in this PCR. Viral RNA and DNA will be extracted from supernatant of NPS samples using the QIAamp cador Pathogen Mini Kit or the Qiacube HT manufactured by Qiagen and PCR reactions will be ran as described elsewhere.[112] PCR for SARS-CoV-2 is not planned because paediatric pneumonia caused by SARS-CoV-2 is generally rare and clinical cases of COVID-19 in adults have been rare in the region. However, if future evidence suggests that SARS-CoV-2 shall be considered as a pathogen of paediatric clinical pneumonia, SARS-CoV-2 PCR may be performed.

► Malarial microscopy: thick blood smear slides will be read blindly by two microscopists and, if discrepant, by a third.

► FBC: FBC will be performed on a Medonic M51 five-part haematology analyser (Boule Diagnostics AB, Domnarvsgatan, Spanga, Sweden). The parameters measured will be white blood cell count, absolute and per cent count of lymphocytes, monocytes, neutrophils, eosinophils, basophils, red blood cell count, platelet count, haemoglobin, mean cell volume, haematocrit, red cell distribution width, mean cell haemoglobin, mean corpuscular haemoglobin concentration and mean platelet volume.

► Multiplex Bead-Based Immunoassay: using Luminex technology will quantify proteins in plasma and will be conducted at the Multiplex Core Facility of the Centre of Translational Immunology ISO9001:2015 laboratory using FlexMAP3D systems.[113] A total of 50 proteins will be included in the assay: (1) proteins identified in our previous study to constitute accurate signatures (eg, HP, interleukin (IL)-10 and IL-18 and tissue inhibitor of metalloproteinases 1, tumour necrosis factor receptor 2 and resistin)[114]; (2) proteins that discriminated the three groups in high-throughput scans conducted previously (eg, bactericidal/permeability-increasing protein, lactotransferrin, granulocyte colony-stimulating factor and matrix metallopeptidase 9); (3) proteins specified in previous studies and listed previously (eg, NGAL, PCT, TRAIL and IP-10), and proteins identified in previous studies listed earlier with prognostic value (eg, angiopoietin).

► Chest radiographs: frontal radiographs will be obtained, and images will be recorded in a computer file. Radiographical findings will be classified as proposed by Cherian *et al*[51] into the presence of consolidation, pleural fluid, other infiltrates or normal. Additionally, the presence of reticular/reticulonodular patterns, cavities, pneumatoceles, lymphadenopathy, abscess, focal fibrosis, pneumothorax, cardiomegaly and foreign body will be noted. Chest X-rays will be read by two clinicians and, if discrepant for consolidation, they will be adjudicated by a third reader, all with training in radiographical interpretation for pneumonia and calibration against WHO standard images. Also, a pulmonologist will read a sample of 10% of X-rays for quality assurance and control.

## Data management

All data will be captured through laptops/desktop computers using an electronic medical record system (EMRS) developed at the MRCG. The only exception will be for follow-up of admitted patients who will be captured on paper forms and later entered in the EMRS. During power outages, paper forms will be completed and, when electricity is back, data will be entered into the EMRS. To ensure data quality, data validation rules have been established in the EMRS that generates alert messages. Additionally, a complete process flow has been established to sequentially activate forms for entry, and a report listing missing variables is generated during entry of forms to assure form completeness.

## Data analysis

Descriptive analysis will be conducted to seek outliers and to assess assumptions for tests and appropriate transformations. Patient severity scores at baseline will be derived based on principal component analysis, including baseline clinical and laboratory variables. These scores will be included in the analysis and used to explore stratification of patients in the study of diagnostic biomarker signatures. Adjustments for potential confounders (eg, overall and cell-specific leucocyte counts, breast feeding and undernutrition) and assessment of heterogeneity across subgroups will be done by incorporating the factor in analysis.

In analysis of objective 1, we will first test previous diagnostic biomarker signatures[5 105] through estimating AUC-ROC, sensitivity and specificity with 95% CI. Using levels of biomarkers included in Valim *et al*[105] and Oved *et al*,[98] we will reproduce signatures using the original models. To discover alternative biomarker signatures, we will start by conducting single biomarker analysis in which we will rank individual markers by pairwise comparisons of the three definitive diagnosis groups (primary analysis) and the two bacterial versus non-bacterial probable diagnosis groups (secondary analysis) and estimation of AUC-ROC with 95% CI.[115] We will also estimate sensitivity and specificity for an optimal cut-off (weighting towards high sensitivity to bacteria) based on comparison of the BA and NBA

definitive diagnosis groups through resampling. Next, to identify composite biomarkers, all markers (including clinical and laboratory tests and severity scores) will be combined and selected through comparison of the three definitive diagnosis groups using logistic regression with elastic net; classification or regression trees depending on the specific outcome; and support vector machine (SVM). In the SVM, non-linear radial basis function kernels will be used and features will be selected through a penalisation and recursive feature elimination.[116 117] We will train models and identify signatures based on the three diagnosis groups, although the focus is in classification of BA and NBA (defined as MA or VI diagnosis) because using three groups, we found that signatures are likely to be more accurate when markers are not monotonically increasing or decreasing across groups. For instance, if a marker is high in BA, low in VI and high in MA, levels of VI with MA are suboptimally averaged when VI and MA are collapsed for training signatures. Within each class of models, tuning parameters will be obtained through a metric of misclassification (eg, deviance or error) estimated in fivefold cross validation. Additionally, models will be selected based on a biased choice of tuning parameters (those determining ≤5 markers). Penalties for missing a bacterial infection will be used. Models will be tested by estimating accuracy to diagnose BA versus non-BA in the definitive diagnosis classification using resampling. Moreover, accuracy will be estimated by comparing BA with NBA probable diagnosis groups.

Analyses of objective 2 will use categorical or continuous surrogates of poor prognosis as outcomes and markers as predictors. Markers will be measured at admission. Signatures will be sought and tested using the same models of objective 1.

To estimate precision, we will report the coefficient of variation (CV) of selected biomarkers in healthy controls. Contaminated cultures will be considered negative. In immunoassays, samples <lowest limit of quantification (LLOQ) or > highest limit of quantification (HLOQ) will be assigned a value below and above the limits of quantification, respectively, through a uniform distribution.

An $\alpha$-significance level of 0.05 will be used. In single-marker analysis, p values and CIs will be adjusted for family-wise error rate through permutation or false discovery rate through Benjamin-Rocheberg, depending on the analysis.[118]

### Sample size justification

We plan to enrol 900 patients to obtain 30 patients with confirmed bacterial disease based on expectations that ~3% of patients will have a positive bacterial culture for objective 1. Also, we anticipate that 30 patients will meet definitive viral or malarial aetiology. If necessary, we will randomly sample patients from the viral and malarial groups. The choice of 30 patients with and 60 patients without bacterial disease was based on feasibility considerations, as was the choice of 20 healthy controls and 160 patients in the analysis of the 'probable' diagnosis

**Table 1** Precision (exact 95% CIs) of sensitivity (sample size of 30) and specificity (sample size of 60) when evaluating diagnostic biomarker signatures in analysis of aim 1

| | 95% CI | |
| Proportion (%) | Sensitivity (%) | Specificity (%) |
|---|---|---|
| 85 | 65 to 94 | 73 to 95 |
| 90 | 73 to 98 | 79 to 90 |
| 95 | 78 to 99 | 86 to 99 |

categories (a random stratified sample from all available subjects). However, with ~90 patients and only 23 patients with bacterial infections, we could previously identify accurate signatures.[105] We anticipate that in objective 1, accuracy of previously identified signatures will be estimated with relatively narrow CIs (table 1). For healthy controls, studying 20 patients will yield a 0.18 CI range for a CV of 0.20 with 0.80 assurance.[119] Analysis of objective 2 will include a larger number of patients than objective 1 and shall yield precise CIs and more stable cross validation-based estimates.

### ETHICS AND DISSEMINATION

Ethics approval has been obtained from the Gambia Government/MRCG Joint Ethics Committee (protocol 1616), and the institutional review board of Boston University Medical Centre (STUDY00000958). We plan to share the findings in peer-reviewed journals.

### PROTOCOL REGISTRATION

The study is registered in clinicaltrials.gov.

**Author affiliations**
[1]Department of Global Health, Boston University School of Public Health, Boston, Massachusetts, USA
[2]Medical Research Council Unit, The Gambia at the London School of Hygiene & Tropical Medicine, Fajara, The Gambia
[3]Center of Translational Immunology, Department of Rheumatology and Clinical Immunology, University Medical Center Utrecht, Utrecht, The Netherlands
[4]Bansang Hospital, Bansang, The Gambia
[5]Basse Hospital, Basse, The Gambia
[6]Hospital Clínic, Universitat de Barcelona, ISGlobal, Barcelona, Spain
[7]Centro de Investigação em Saúde de Manhiça (CISM), Maputo, Mozambique
[8]Luminex Corp, Austin, Texas, USA
[9]Department of Medicine, Harvard Medical School, Boston, Massachusetts, USA
[10]Division of Pulmonary and Critical Care Medicine, Brigham and Women's Hospital, Boston, Massachusetts, USA
[11]Independent Researcher, Boston, Massachusetts, USA
[12]Division of Infectious Diseases, University of Louisville, Louisville, Kentucky, USA
[13]Vascular Biology Program, Children's Hospital Boston, Boston, Massachusetts, USA
[14]Department of Surgery, Harvard Medical School, Boston, Massachusetts, USA
[15]Disease Elimination and Control, Medical Research Council Unit, Fajara, Gambia
[16]London School of Hygiene & Tropical Medicine, London, UK
[17]Boston University School of Public Health, Boston, Massachusetts, USA
[18]Department of Disease Control, Faculty of Infectious and Tropical Diseases, London School of Hygiene & Tropical Medicine, London, UK

# Open access

**Acknowledgements** We thank the contributions of the clinical, laboratory and data management staff of Medical Research Council Unit The Gambia to the final version of this study protocol. Furthermore, we are grateful to Dr Terrie Taylor for her invaluable support, edits and guidance to this work.

**Contributors** The study concept and design were conceived by CV, GAM and PLH with contributions from EFK, WdJ, AAD, RCW, JR, MAM, UDA and QB. Acquisition of clinical data were defined by CV, GAM, PLH, YAO, YSI, RS, SG, SK and AJ. Acquisition of laboratory data was defined by CV, RS, EFK, WdJ, RCW, MAM and GAM. The analytical plan was conceived by CV. Participant enrolment and follow-up will be conducted by YAO and YSI under direct supervision of GAM and oversight from PLH, SK, AJ and QB. Sample processing and diagnostic laboratory assays will be supervised by RS and data management by SG. All on-the-ground operations in the Gambia will be directly supervised by GM and overseen by CV. QB will be one of the clinicians assigning participants to one of the probable diagnostic groups. The multiplex immunoassay to quantify proteins will be conducted by EFK with inputs from CV, WdJ, RCW, JR and MAM. Analysis will be performed by CV. CV prepared the first draft of the manuscript with inputs from GAM, PLH, YAO, YSI, RS and SG. CV, YAO, YAI, RS, SG, EFK, SK, AJ, QB, WDJ, AAD, RCW, JR, MAM, UDA, PLH and GAM provided edits, reviewed and approved the final version of the manuscript, providing substantial intellectual contributions.

**Funding** This study will be supported by the National Institutes of Health of the USA (R21AI140258) and will leverage resources from the project entitled 'Will the Ongoing Use of a Two-Dose, Rather Than Three-Dose Schedule of Pneumococcal Conjugate Vaccine, Have Similar Impact in Rural Gambia' study funded by the Joint Global Health Trials scheme (MRC UK, Welcome, UK AID, UK National Institute for Health Research) and the Bill and Melinda Gates Foundation.

**Competing interests** None declared.

**Patient and public involvement** Patients and/or the public were not involved in the design, conduct, reporting or dissemination plans of this research.

**Patient consent for publication** Not applicable.

**Provenance and peer review** Not commissioned; externally peer reviewed.

**ORCID iDs**
Clarissa Valim http://orcid.org/0000-0001-9621-203X
Grant A Mackenzie http://orcid.org/0000-0002-4994-2627

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
