## [Reviewer comments · BMJ Open]

ARTICLE DETAILS

TITLE (PROVISIONAL)	A Study Protocol for an Observational Study Seeking Diagnostic and Prognostic Biomarkers for Childhood Bacterial Pneumonia in Sub-Saharan Africa
AUTHORS	Valim, Clarissa; Olatunji, Yekin; Isa, Yasir; Salaudeen, Rasheed; Golam, Sarwar; Knol, Edward; Kanyi, Sheriffo; Jammeh, Abdoulie; Bassat, Quique; deJager, Wilco; Diaz, Alejandro; Wiegand, Roger; Ramirez, Julio; Moses, Marsha; D'Alessandro, Umberto; Hibberd, Patricia; Mackenzie, Grant

VERSION 1 – REVIEW

REVIEWER	Mejbah Bhuiyan The University of Western Australia, Division of Paediatrics, School of Medicine, Faculty of Health and Medical Sciences
REVIEW RETURNED	22-Jan-2021

GENERAL COMMENTS	Title: A Study Protocol for an Observational Study Seeking Diagnostic and Prognostic Biomarkers for Childhood Bacterial Pneumonia in Sub-Saharan Africa Thanks for the opportunity to review the study protocol. The study aims look to have huge implication to respond to the global antimicrobial resistance crisis. However, sadly, the paper needs a lot of further work to help understand the study activities clearly. Small sample size is the biggest challenge to meet the primary objective and it is recommended the researchers think carefully before implementing the study knowing the chance to reach valid conclusion is really low. My specific comments are below: Abstract: • There are accurate methods like blood culture and molecular methods for distinguishing bacterial pneumonia from malaria and . It is not clear what the researchers wanted to claim – please rephrase.• Which proteins will be measured? It is recommended to mention few primary proteins if not all.• A brief description of statistical methods is required. Introduction: • Please check referencen#1, the context of the first statement may not be supported by ref#1. Additional reference is suggested.• It is recommended to mention main outcome for “publications on biomarkers for the diagnosis of bacterial infections” rather than lumping >30 references together, which is very unusual.• While PCR needs specialised resources but their accuracy for virus detection is high, of course it depends on the samples been used. Not completely agree with the authors statement.• It is important to the outcome of references than citing the reference to clarify what had been done. This would also identify the
--

	gaps and strengthen the rationale of study. What was the biomarkers with promising results ref #94, #95?  • The introduction section, overall, has been unnecessarily expanded and should be condensed with critical information only. Methods: The methods were described in detail but difficult to follow. The objectives were clear but there are some trouble in understanding the outcomes.  • Objective 1 contains two analysis : primary and secondary. It is not clear why one objective contains two type of analysis. And also it was not clear what and how these two analysis would achieve. • Objective 2 was termed as Poor prognosis – very different from what had been stated as objective 2 “Explore prognostic biomarker signatures in patients with clinical pneumonia caused by any pathogen”. It was expected to have discussion about prognostic biomarkers signature to support identification of clinical pneumonia but there were discussion about different prognosis. • A reference is required for WHO criteria used in this study for participant identification. • Recruitment and follow up should be followed by Fig 1 not Fig 2. • Specimen collection and testing are appropriate. While the laboratory methods were previously used, it would be worth to describe the process in brief for the readers. • Specimen transportation plan from site to BUSPH for immunoassay deserves mentioning. • A lot of abbreviation throughout the paper (CTI?UMC, TIMP-1, BPI, EPHB2 LTF...) that needs to be mentioned first then abbreviated. • The data analysis part is very rich, suggesting the capacity of the authors in doing statistical analysis. However, it was indeed difficult to follow the analysis to meet the objective set. • The sample size section needs clarity. What are the available data that made the authors confident about the precision estimates? These data were not presented with reference, so the estimates could not be assessed. • The method needs proper chronology like participants, eligibility criteria, f/u and laboratory procedure, data analysis etc should come before outcomes. • The primary shortfalls of this study are:  o Small sample size. It is very unlikely that with few patients like 20-30, the study would able to reach its primary outcome. While it was mentioned by the authors as limitation but if the study is highly likely failing to meet the primary objective for a limitation, the researchers should be more careful how to solve the factor first before conducting the study. o While the study criticises in the background section that the current diagnostic methods are not optimum to provide results promptly and also needs expertise(like for PCR) that many LMICs don't have, the immunoassay they are proposing even require high quality expertise than basic PCR. This will definitely not help the LMICs. How the authors want to defend the use of methods require more expertise (immunoassay) given the rationale they set for this study? Finally, it was not really unclear from the paper how and what are likely prognostic factors that could help in developing POCs for bacterial clinical pneumonia in LMICs and add new evidence to existing literature.
--	---

REVIEWER	Massimiliano Don
-----------------	------------------

	Sant'Antonio General Hospital
REVIEW RETURNED	11-Feb-2021

GENERAL COMMENTS	The study protocol by Valim et al. is clear and well articulated. In particular, Abstract and Introduction are well structured; the latter is based on updated references and includes two objectives, which respectively focus on the identification of diagnostic and prognostic biomarkers for childhood bacterial pneumonia in a developing country. Minor comments for the Methods and Analysis chapter are the following:  • Some specifics about HIV diagnosis are missing (pg 13 and 16 of 35) • Are research nurses really skilled for doing and interpret a correct physical examination? (pg 13 of 35) • An abbreviations' list is missing; • Does coronavirus test include SARS-CoV-2 test? (pg. 17 on 35) • Better specify what full blood cell count means (pg. 17 on 35)
---

VERSION 1 – AUTHOR RESPONSE

Reviewer: 1

Mr. Mejbah Bhuiyan, The University of Western Australia

Comments to the Author:

Title: A Study Protocol for an Observational Study Seeking Diagnostic and Prognostic Biomarkers for Childhood Bacterial Pneumonia in Sub-Saharan Africa

Thanks for the opportunity to review the study protocol. The study aims look to have huge implication to respond to the global antimicrobial resistance crisis. However, sadly, the paper needs a lot of further work to help understand the study activities clearly. Small sample size is the biggest challenge to meet the primary objective and it is recommended the researchers think carefully before implementing the study knowing the chance to reach valid conclusion is really low.

My specific comments are below:

Abstract:

- There are accurate methods like blood culture and molecular methods for distinguishing bacterial pneumonia from malaria and . It is not clear what the researchers wanted to claim – please rephrase.

RESPONSE: Blood culture is highly specific but not sensitive and PCR has also limited sensitivity. We clarified that in the Abstract.

- Which proteins will be measured? It is recommended to mention few primary proteins if not all.

RESPONSE: Given the limit in the Abstract word count, we could not provide names of proteins in the Abstract. However, to address the reviewer concern, we added some additional target proteins to the ones previously stated in the body of the manuscript. We cannot provide at this point a the full list of proteins to be quantified because knowledge on potential markers evolves rapidly and we may need to add or remove some proteins to the list we currently plan to measure.

- A brief description of statistical methods is required.

RESPONSE: We incorporated a brief description in the Methods and Analysis of the manuscript Abstract

Introduction:

- Please check referencen#1, the context of the first statement may not be supported by ref#1. Additional reference is suggested.

RESPONSE: We moved the original reference # 1 to the middle of the sentence and added (now reference #2) to the end of the sentence, supporting the fact that *Plasmodium falciparum* malaria can lead to a pneumonia-like disease

- It is recommended to mention main outcome for “publications on biomarkers for the diagnosis of bacterial infections” rather than lumping >30 references together, which is very unusual.

RESPONSE: We understand the reviewer’s concern. However, the main outcome of all those studies is stated, i.e., AUC-ROC was < 0.80. We thought was important to honor the author of all these studies and add them to the reference list. However, to detail their outcomes would substantially increase the word counting.

- While PCR needs specialised resources but their accuracy for virus detection is high, of course it depends on the samples been used. Not completely agree with the authors statement.

RESPONSE: To our knowledge bacterial PCR has limited sensitivity. Indeed viral PCR has a high sensitivity to detect viruses in nasopharyngeal swabs. However, detection of virus in nasopharyngeal swabs does not establish a diagnosis of viral pneumonia because viruses are often detected in healthy children. Detection of virus in pleural fluid liquid or lung aspirate does establish a diagnosis of viral pneumonia but these samples are only obtainable in specific cases and are based on invasive procedures. We added a sentence to clarify that in the Introduction.

- It is important to the outcome of references than citing the reference to clarify what had been done. This would also identify the gaps and strengthen the rationale of study. What was the biomarkers with promising results ref #94, #95?

RESPONSE: A sentence was added to briefly specify the result of the original references # 94 and #95 (currently 96 and 97)

- The introduction section, overall, has been unnecessarily expanded and should be condensed with critical information only.

RESPONSE: It was unclear to us what the reviewer considered non-essential information. We deleted two sentences from the Introduction that we assumed were considered unnecessary by the reviewer.

Methods:

The methods were described in detail but difficult to follow.

The objectives were clear but there are some trouble in understanding the outcomes.

- Objective 1 contains two analysis : primary and secondary. It is not clear why one objective

contains two type of analysis. And also it was not clear what and how these two analysis would achieve.

RESPONSE: We added a sentence under “Objective I” to explain the importance of the primary and secondary outcome. The two analysis play the role of primary and secondary outcomes in a clinical research study. Briefly, the primary outcome (definitive classification criteria) requires that patients have bacteremia. The presence of bacteremia may affect inflammatory markers. Since the purpose of a point-of-care test based on the identified combination of inflammatory proteins would be to diagnose occurrence of bacterial infections in patients with and without bacteremia, assessing proteins in patients with the “probable diagnosis” (most without bacteremia) would support the generalizability of the identified signatures. A description of how the analysis will be achieved is explained under Data Analysis.

- Objective 2 was termed as Poor prognosis – very different from what had been stated as objective 2 “Explore prognostic biomarker signatures in patients with clinical pneumonia caused by any pathogen”. It was expected to have discussion about prognostic biomarkers signature to support identification of clinical pneumonia but there were discussion about different prognosis.

RESPONSE: The title Poor Prognosis is the heading for the “**Outcomes**” of Objective II. The Objective II (titled “Exploring prognostic signatures”) is based on the outcome titled “Poor Prognosis”.

- A reference is required for WHO criteria used in this study for participant identification.

RESPONSE: The following reference has been added, ‘Integrated Management of Childhood Illness – Chart Booklet, World Health Organization, Geneva, March 2014, www.who.int/maternal_child_adolescent/en, ISBN 9789241506823’

- Recruitment and follow up should be followed by Fig 1 not Fig 2.

RESPONSE: We would like to thank the Reviewer for catching this mistake. A correction was made and the paragraph is now making reference to Fig. 1 and not Fig. 2

- Specimen collection and testing are appropriate. While the laboratory methods were previously used, it would be worth to describe the process in brief for the readers.

RESPONSE: We added the parameters that will be estimated in the full blood cell count. We apologize but we did not understand which other laboratory methods the reviewer wanted to have explained in more detail. Because these are very standard assays and given concerns with the manuscript length, we did not add any further description.

- Specimen transportation plan from site to BUSPH for immunoassay deserves mentioning.

RESPONSE: We added a sentence at the end of Recruitment and Follow-up. The samples will be shipped to the Center of Translational Immunology at the Medical Center of the University of Utrecht through a temperature monitored shipping company that replenishes dry ice, one of our partner institutions, and not to BUSPH.

- A lot of abbreviation throughout the paper (CTI?UMC, TIMP-1, BPI, EPHB2 LTF...) that needs to be mentioned first then abbreviated.

RESPONSE: We deleted the acronym CTI/UMC, added an abbreviation list for the proteins, and wrote down the acronym when it first appeared.

- The data analysis part is very rich, suggesting the capacity of the authors in doing statistical analysis. However, it was indeed difficult to follow the analysis to meet the objective set.

RESPONSE: We thank the reviewer for the kind comment.

- The sample size section needs clarity. What are the available data that made the authors confident about the precision estimates? These data were not presented with reference, so the estimates could not be assessed.

RESPONSE: Sample size for estimation of sensitivity and specificity is based on width of 95% confidence intervals. Table 1 presents confidence intervals for our target sensitivity and specificity of 85% and higher. We also added a reference to the Sample Size section.

- The method needs proper chronology like participants, eligibility criteria, f/u and laboratory procedure, data analysis etc should come before outcomes.

RESPONSE: The Analytical Plan and laboratory methods depend on the outcomes. Because of that, publications, protocols, and grant applications oftentimes present the Outcomes before these sessions. Although we understand the reviewer preference, we chose to not change the order of the subsections of the Methods. We hope that is acceptable for the reviewer.

- The primary shortfalls of this study are:
 - o Small sample size. It is very unlikely that with few patients like 20-30, the study would be able to reach its primary outcome. While it was mentioned by the authors as a limitation but if the study is highly likely failing to meet the primary objective for a limitation, the researchers should be more careful how to solve the factor first before conducting the study.

RESPONSE: We have added a sentence remarking that the previous study we conducted and published in the Blue Journal (American Journal of Respiratory and Critical Care Medicine) had only 23 bacterial patients. The proposed study will include more patients than the previous study, i.e., 30 patients in each of the three comparison groups based on the definitive criteria and 160 patients based on the probable criteria. Although we understand the reviewer's concern that this sample size is limited, given our funding and the low sensitivity of bacterial cultures it would be unfeasible for us to study a larger sample size. We believe that, as in the previous study, by using a rigorous patient classification the limited sample size will yield important conclusions that will allow us to carry out a large and definitive study. Several other biomarkers studies in different applications have yielded important conclusions with a sample size more limited than ours.

- o While the study criticises in the background section that the current diagnostic methods are not optimum to provide results promptly and also needs expertise (like for PCR) that many LMICs don't have, the immunoassay they are proposing even requires high quality expertise than basic PCR. This will definitely not help the LMICs. How do the authors want to defend the use of methods that require more expertise (immunoassay) given the rationale they set for this study?

RESPONSE: As specified in the introduction, we aim to incorporate the selected proteins in a point-of-care diagnosis. Currently, there are different technologies to multiplex proteins in point-of-care that can be used for inexpensive diagnostics. Examples of such diagnostics are applied to flu virus and malaria rapid diagnostic tests based on lateral flow. Development and testing of such diagnosis will be our next step.

Finally, it was not really unclear from the paper how and what are likely prognostic factors that could help in developing POCs for bacterial clinical pneumonia in LMICs and add new evidence to existing literature.

RESPONSE: We did not fully understand the reviewer comment. The specific prognostic factors that will be sought are combinations of inflammatory proteins. The outcomes are detailed under "Outcomes".

Reviewer: 2

Dr. Massimiliano Don, Sant'Antonio General Hospital

Comments to the Author:

The study protocol by Valim et al. is clear and well articulated.

In particular, Abstract and Introduction are well structured; the latter is based on updated references and includes two objectives, which respectively focus on the identification of diagnostic and prognostic biomarkers for childhood bacterial pneumonia in a developing country.

Minor comments for the Methods and Analysis chapter are the following:

- Some specifics about HIV diagnosis are missing (pg 13 and 16 of 35)

RESPONSE: Sentences have been added describing that, 'We will follow local clinical practice and HIV testing will not be routinely conducted. HIV prevalence is approximately 1% in antenatal care in The Gambia.'

- Are research nurses really skilled for doing and interpret a correct physical examination? (pg 13 of 35)

RESPONSE: The full clinical assessment after patients have been enrolled is based on experienced Research Clinicians, all pediatricians. The Research Nurses only conduct clinical evaluation for eligibility assessment. These nurses are senior nurses that have participated in other pneumonia studies carried out in the area. The same nurses are performing eligibility assessment for a large pneumococcal vaccine randomized trial in the area and are under constant supervision. We added a few words explaining that in the "Recruitment and follow-up" subsection.

- An abbreviations' list is missing;

RESPONSE: We added an abbreviation list and also defined in the text acronyms that were missing a definition.

- Does coronavirus test include SARS-CoV-2 test? (pg. 17 on 35)

RESPONSE: As of now, it does not. We added a statement specifying that in the description of the “Viral PCR of NPS”. We are not planning to perform SARS-CoV-2 PCR considering: a) SARS-CoV-2 has rarely occurred in the area of the study; b) cases of severe pneumonia in children caused by SARS-CoV-2 are rare; c) funding limitation. However, if evidence emerges that SARS-CoV-2 shall be considered in the etiology of pediatric clinical pneumonia in the region, we may include PCR for detection of SARS-CoV-2 in the panel of diagnostic methods

- Better specify what full blood cell count means (pg. 17 on 35)

RESPONSE: We added the parameters that will be measured by the full blood cell count to the paragraph “Full blood cell count (FBC)”.